# Genetic and molecular characterization of multicomponent resistance of *Pseudomonas* against allicin

Jan Borlinghaus[1] , Anthony Bolger[2], Christina Schier[1], Alexander Vogel[2], Björn Usadel[2] , Martin CH Gruhlke[1], Alan J Slusarenko[1]

**The common foodstuff garlic produces the potent antibiotic defense substance allicin after tissue damage. Allicin is a redox toxin that oxidizes glutathione and cellular proteins and makes garlic a highly hostile environment for non-adapted microbes. Genomic clones from a highly allicin-resistant *Pseudomonas fluorescens* (*Pf*AR-1), which was isolated from garlic, conferred allicin resistance to *Pseudomonas syringae* and even to *Escherichia coli*. Resistance-conferring genes had redox-related functions and were on core fragments from three similar genomic islands identified by sequencing and in silico analysis. Transposon mutagenesis and overexpression analyses revealed the contribution of individual candidate genes to allicin resistance. Taken together, our data define a multicomponent resistance mechanism against allicin in *Pf*AR-1, achieved through horizontal gene transfer.**

## Introduction

Plants produce a vast array of secondary metabolites, many of which are involved in defense against microbes, resulting in a dynamic co-evolutionary arms race in the interaction between plants and their associated microorganisms (Burdon & Thrall, 2009). Plants provide habitats for commensal and pathogenic organisms and generally it is assumed that microorganisms found in association with a given plant host are adapted to that ecological niche as part of the microbiota. Adaptation is the process that tailors organisms to a particular environment and enhances their evolutionary fitness, and the organosulfur compounds produced by garlic (*Allium sativum* L.) provide an important example of this scenario. The potent antibacterial activity of garlic is mainly due to diallylthiosulfinate (allicin) (Cavallito & Bailey, 1944; Cavallito et al, 1944). Allicin, which is responsible for the typical odor of freshly crushed garlic, is formed by the action of alliin lyase (E.C.4.4.1.4) on alliin (*S*-allyl-L-cysteine sulfoxide) when the enzyme and substrate mix after damage to garlic tissues. The reaction

proceeds rapidly, and alliin conversion to allicin is ~97% complete after 30 s at 23°C (Lawson & Hughes, 1992). Alliin lyase is one of the most prevalent soluble proteins found in garlic bulbs and leaves, and a single clove of ~10 g fresh weight can liberate up to 5 mg of allicin (Lawson et al, 1991a; Block, 2010), revealing a major investment of plant resources into this defense system (Van Damme et al, 1992; Smeets et al, 1997; Borlinghaus et al, 2014).

Allicin has multiple sites of action and is a concentration-dependent biocide, active against bacteria, fungi, oomycetes, and mammalian cells (Borlinghaus et al, 2014). Allicin is an electrophile that oxidizes thiols, or more precisely the thiolate ion, in a modified thiol–disulfide exchange reaction, producing *S*-allylmercapto disulfides (Miron et al, 2000; Müller et al, 2016). Cellular targets include accessible cysteines in proteins, and the cellular redox buffer glutathione (GSH). In this way, allicin can inhibit essential enzymes (Wills, 1956) and shift the cell redox balance (Gruhlke et al, 2010), causing oxidative stress. Indeed, at sublethal doses, allicin was shown to activate the Yap1 transcription factor that coordinates the protective oxidative stress response in yeast (Gruhlke et al, 2017). There are indications that the allicin target and cellular redox buffer glutathione (GSH) plays a central role in enabling cells to resist the effects of allicin (Gruhlke et al, 2010, 2017; Leontiev et al, 2018). Allicin reversibly *S*-thioallylates a range of proteins in bacteria and human cells which can lead to loss of function of essential enzymes (Müller et al, 2016; Chi et al, 2019; Gruhlke et al, 2019; Loi et al, 2019; Wüllner et al, 2019).

Sensitivity to allicin varies between different bacteria, but the basis for this is unknown (Reiter et al, 2017). We isolated a highly allicin-resistant *Pseudomonas fluorescens*, *Pf*AR-1, from a clove of garlic. How resistance against allicin might be conditioned in *Pf*AR-1 and how it arose are intriguing questions. One possibility for the acquisition of multicomponent resistance is horizontal gene transfer (HGT), that is, the sharing of genetic material between organisms that are not in a parent–offspring relationship. Large, chromosomally integrated regions obtained by HGT are referred to as genomic islands (GIs), and these are known to expand the ecological niches of their host bacteria for complex and competitive environments (Soucy et al, 2015). GIs generally show a different average GC content and codon usage to the rest of the

[1]Department of Plant Physiology, Rheinisch-Westfälische Technische Hochschule Aachen (RWTH Aachen University), Aachen, Germany   [2]Department of Botany, Rheinisch-Westfälische Technische Hochschule Aachen (RWTH Aachen University), Aachen, Germany

Correspondence: jan.borlinghaus@rwth-aaachen.de; alan.slusarenko@bio3.rwth-aachen.de

genome. HGT is a widely recognized mechanism for adaptation in bacteria, and microbial antibiotic resistance and pathogenicity traits are often associated with HGT (MacLean & San Millan, 2019).

In the study reported here, we isolated a highly allicin-resistant bacterium from its ecological niche on garlic, an environment hostile to non-adapted microorganisms, and we used a shotgun genomic cloning strategy to functionally identify genes conferring allicin resistance. The annotated functions of resistance-conferring genes throw light on the complex molecular mechanisms of resistance of PfAR-1 to allicin, a redox toxin which has multiple effects within cells. This functional approach was complemented by whole-genome sequencing which revealed unique genomic features in comparison with other Pseudomonads. Both approaches independently identified the same sets of genes, validating the strategy. The multiple copies of the genes conferring allicin resistance, gained by horizontal transfer and duplication events, emphasize the evolutionary investment associated with allicin resistance in PfAR-1 that presumably enables it to exploit garlic as an environmental niche.

## Results

### An allicin-resistant *P. fluorescens* from garlic

We reasoned that if allicin-resistant bacteria were to be found in nature, it would likely be in association with garlic cloves. Therefore, the degree of allicin resistance of bacteria isolated from garlic bulbs was tested in a Petri plate agar diffusion test with bacteria-seeded agar. An isolate that was able to grow right up to the allicin solution was detected. In comparison, allicin-sensitive *Escherichia coli* DH5α and *Pseudomonas syringae* pv. *phaseolicola* Ps4612 showed large inhibition zones (Fig 1A). The allicin-resistant isolate was identified by Sanger sequencing of the ribosomal internal transcribed spacer as *P. fluorescens* and was named PfAR-1 (*P. fluorescens* Allicin Resistant-1).

PfAR-1 genomic clones were shotgun electroporated into cells of highly allicin-sensitive Ps4612. In all, $1.92 \times 10^8$ clones were screened, giving ~33× library coverage. Resistant recipients were selected on allicin-containing medium, and eight resistant transformants were confirmed in streak tests (Fig 1B). Restriction analysis revealed that the resistance-conferring PfAR-1 clones were all ~10 kb in size.

In both *E. coli* and Ps1448A, it was found that PfAR-1 clones conferred resistance to allicin but not to the other oxidants tested (Fig 1C). The degree of allicin resistance conferred by genomic clones 1 and 5 was similar, but clone 8 was less effective than the other clones (Fig 1C). The different oxidizing agents tested cause different stresses in cells. Thus, allicin *S*-thioallylates -SH groups, which is a reversible thiol modification similar to glutathiolation (Gruhlke et al, 2019). Whereas $H_2O_2$ is a reactive oxygen species that reacts poorly with -SH groups and is largely removed from cells by peroxiredoxins (Winterbourn & Hampton, 2008; Poole, 2015), CHPO causes lipid peroxidation (Halliwell & Gutteridge, 2015), and *N*-ethylmaleimide (NEM), although oxidizing -SH groups, does so irreversibly. The results show that the clones do not confer resistance to oxidative stress in general, but rather to the type of oxidative stress caused by allicin in particular.

### In silico analysis of the *Pf*AR-1 genome

The PfAR-1 genome was sequenced using a combined Illumina and Pacific Biosciences data set and assembled into a single chromosome, as described in the Materials and Methods section. The genome size was determined to comprise 6,251,798 bp and had an overall GC content of 59.7%. A total of 5,406 putative protein-coding sequences, in addition to 73 tRNAs and 6 rRNA clusters, were detected. With an average nucleotide identity (ANI) of 85.94% (determined with OrthoANI software, [Lee et al, 2016]), the closest relative to PfAR-1 in the databases was *P. fluorescens* reference strain Pf0-1, supporting the prior internal transcribed spacer–based identification.

Sanger sequencing of the clone ends was used to identify the origin of the clones within the sequenced PfAR-1 genome. This revealed that clones 1 and 8 had unique origins, whereas clones 2–7 were identical. Thus, three relatively compact allicin resistance–conferring genomic regions had been identified. Genes carried on the clones had preponderantly redox-related functions (Fig 2A and B and Table 1), which fits with allicin's redox toxin mode of action. The overall arrangement of the genes was highly conserved among the clones. Clones 1–7 contained two sets of genes, both of which were conserved in the direction of transcription: *osmC*, *sdr*, *tetR*, *dsbA*, and *trx*; and *ahpD*, *oye*, *4-ot*, *kefF*, and *kefC*, respectively (Fig 2C). Clone 8, which conferred slightly less allicin resistance than the other clones (Fig 1C), lacked *ahpD* and *oye* genes. The *kefF* and *kefC* genes are part of a glutathione-regulated K$^+$ efflux/H$^+$ influx system and are classified as transporters, although they too are regulated by cellular glutathione and, thus, are redox-dependent.

*P. fluorescens* Pf0-1 is PfAR-1's closest sequenced relative. Nonetheless, dot matrix alignment of the Pf0-1 and PfAR-1 genomes revealed substantial differences. The PfAR-1 chromosome had a central inverted region with respect to Pf0-1, and three large GIs with lower GC content (<55%), which were absent in Pf0-1 (Fig 3A and B). The combination of low GC content and absence from the genome of a near-relative suggests that these regions might have arisen by HGT. Further analysis revealed that each of the three GIs (GI1, GI2, and GI3) contained a highly similar region, which we labeled repeat RE1, RE2, and RE3, respectively. The genes within these repeat regions had many annotations in common and a syntenic organization (Table S1), suggesting a shared origin.

It is unusual for multiple copies of genes to be maintained in bacteria without a clear selective advantage because of the genomic instability that arises through homologous recombination leading to genome rearrangements and loss of essential interim sequences (Rocha, 2003). Intriguingly, the allicin resistance–conferring clones found in the functional analysis originated within these three repeat regions (Fig 3B–E), suggesting that the selective advantage may be, in fact, the increased allicin resistance. Possible origins for the putative HGT regions into the PfAR-1 genome were investigated more closely.

Genes on RE1 and RE2 appeared more closely related to each other than to those on RE3 from both a gene commonality (Jaccard similarity of 90% for RE1 versus RE2, compared with 54.2% for RE1 versus RE3, and 50% for RE2 versus RE3) and amino acid–similarity perspective (97.5%, 87.1%, and 87.2%, respectively). This suggested that RE1 and RE2 originated from a more recent sequence duplication and that RE3 resulted from an earlier duplication event from the common ancestor of RE1 and RE2. To determine the distribution of similar REs within the *Pseudomonas* genus, we arranged these sequences to form a bait set, and compared this against all 3,347 available *Pseudomonas* genomes. Similar regions to the bait were detected in eight of the complete genomes, of which six were from plant-pathogenic or plant-associated

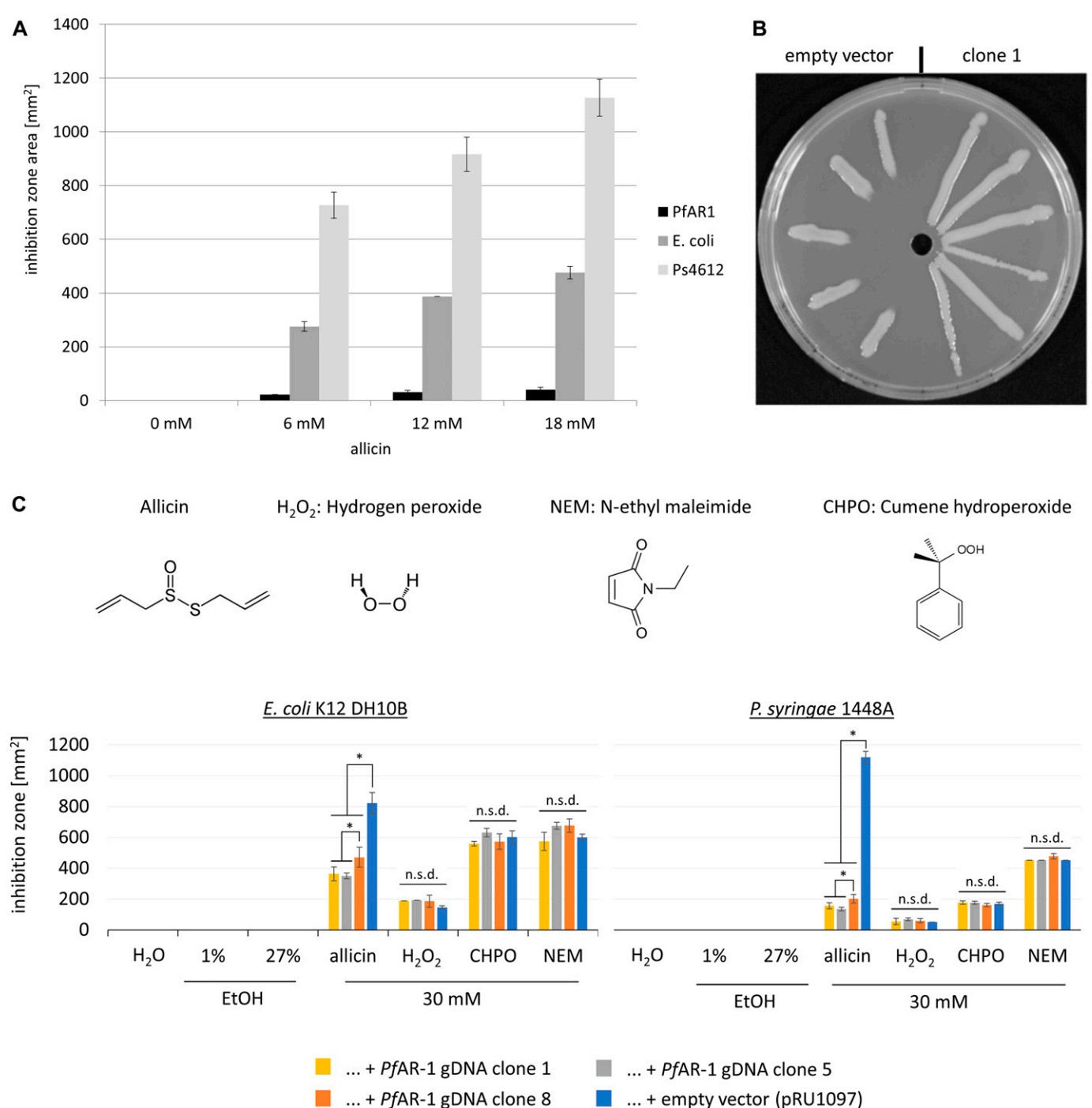

**Figure 1. PfAR-1 is highly allicin-resistant and discrete genomic clones conferred allicin resistance.**
**(A)** Comparison of the sensitivity of PfAR-1, E. coli DH5α, and P. syringae Ps4612 to allicin. The area of the inhibition zones in an agar diffusion test is shown for 40 μl of 0–18 mM allicin applied centrally to a well in the seeded agar medium. n = 3 technical replicates. **(B)** Allicin resistance was conferred by genomic clones from PfAR-1 electoporated into Ps4612. On the left half of the Petri plate, Ps4612 cells contain empty vector and on the right half Ps4612 cells were transformed with vector containing genomic clone 1. The central wells contained 30 μl of 32 mM allicin solution. **(C)** 40 μl of 30 mM allicin, $H_2O_2$, N-ethylmaleimide (NEM), or cumene hydroperoxide (CHPO) were applied to wells cut in agar with a surface lawn of dispersed bacteria. Ethanol was used as a solvent for NEM and CHPO and 1% and 27% ethanol, respectively, were included as controls. Areas of the inhibition zones are shown for the recipients containing genomic clones 1, 5, 8, or the empty vector (n = 3 or more technical replicates, * = P < 0.05, Holm–Sidak method for all pairwise comparison). n.s.d., no significant difference. Each experiment was performed twice. Error bars indicate standard deviation.

pseudomonads. Matching regions were also found in 56 of the draft genomes, eight of which showed two copies of the region. One of the draft genomes had the matching region split across two

contigs, although this was presumably due to incomplete assembly, rather than representing a biological signal. These similar regions ranged from effectively complete, with hits from all 26 bait groups, to

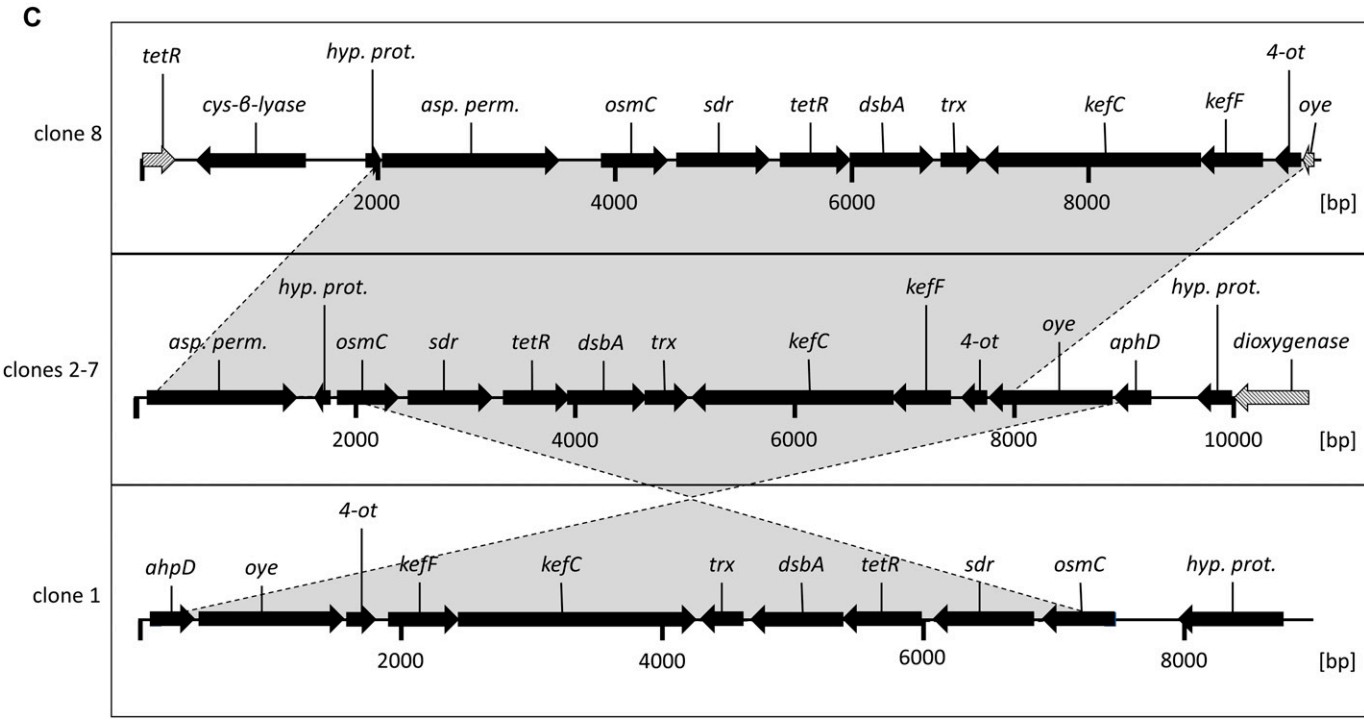

**Figure 2. Characteristics of the allicin resistance–conferring *Pf*AR-1 genomic clones.**
**(A)** Venn diagram showing congruent genes. Annotation is based on protein domains and corresponding families, proteins with no similarities are labelled unknown. **(B)** Congruent genes grouped by function. **(C)** Arrows show the direction of transcription. Grey-shaded arrows in clones 8 and 2–7 represent truncated genes (the genes are annotated fully in Table 1).

highly divergent with only five of the bait groups found. Of the 56 partial genome sequences, 37 were from plant-pathogenic or plant-associated bacteria (Table S2).

Expecting the codon usage of a horizontally transferred gene region to resemble the donor species rather than the current

host, we performed a codon usage analysis to complement the bait sequence analysis described above. For this, we compared the full *Pf*AR-1 genome, the three RE regions, the 3,347 other available *Pseudomonas* genomes, and eight representative non-*Pseudomonas* Gammaproteobacteria. The results were

**Table 1.** Congruent set of genes identified in the genomic clones 1–7 that conferred allicin resistance to *E. coli* K12 DH10B and *P. syringae* pv. *phaseolicola* 4612.

| *Pf*AR-1 genes[a] | Reported function in other bacteria | References |
|---|---|---|
| Alkylhydroperoxidase (*ahpD*) | Part of the carboxymuconolate decarboxylase family. NADH-dependent AhpD/AhpC system confers oxidative stress resistance in *Mycobacterium tuberculosis*. | Bryk et al (2002) and Koshkin et al (2003) |
| Old yellow enzyme (*oye*) | OYE protein family contains a diverse set of NADPH-dependent dehydrogenases that reduce $\alpha,\beta$ unsaturated aldehydes and ketones. OYE was reported to be part of the oxidative stress response in yeast and in *Bacillus*. | Stott et al (1993), Vaz et al (1995), Fitzpatrick et al (2003), and Trotter et al (2006) |
| 4-oxalocrotonate tautomerase (*4-ot*) | 4-OT converts 2-hydroxymuconate to the $\alpha,\beta$-unsaturated ketone 2-oxo-3-hexendioate. | Whitman et al (1991) and Whitman (2002) |
| Glutathione-regulated potassium-efflux system protein F (*kefF*) | KefF is a cytoplasmic regulator of KefC. KefF is activated by glutathione-adducts and subsequently activates KefC. | Meury and Kepes (1982), Elmore et al (1990), Douglas et al (1991), Munro et al (1991), Ferguson et al (1997), Miller et al (2000), and Lyngberg et al (2011) |
| Glutathione-regulated potassium-efflux system protein C (*kefC*) | KefC is a proton import/potassium export antiporter. KefC activity is tightly regulated by glutathione and KefF. Active KefC confers resistance against electrophiles such as *N*-ethylmaleimide in *E. coli*. | |
| Thioredoxin (*trx*) | Trx are dithiol-disulfide oxidoreductases that help to maintain the thiol groups of proteins in a reduced state | Holmgren (2000) |
| Disulfide bond protein A (*dsbA*)/frnE-like | DsbA *in E. coli* is responsible for introduction of disulfide bonds in nascent polypeptide chains in the periplasmic space. Other Dsb members show chaperone-like functions. FrnE is a member of the DsbA family and was reported to confer oxidative stress resistance in *Deionococcus radiodurans*. | Bardwell et al (1991), Kamitani et al (1992), and Khairnar et al (2013) |
| Transcriptional regulator (*tetR*) | Transcriptional repressors widely distributed among different bacteria. | Ramos et al (2005) |
| Short chain dehydrogenase (*sdr*) | The family contains dehydratases, decarboxylases or simple oxidoreductases. | Kavanagh et al (2008) |
| Osmotically inducible protein C (*osmC*) | The family contains peroxiredoxins which play a role in oxidative stress defense. OsmC confers resistance to organic hydroperoxides such as cumene hydroperoxide in *E. coli*. | Lesniak et al (2003) |

[a]Annotation is based on protein domains and corresponding protein families.

plotted using principal component analysis and are shown in Fig S1.

The first principal component, which accounts for almost 78% of the variation, seems to reflect the GC content, ranging from *Acinetobacter baumannii* with a GC content of 38.9% on one extreme to *Rugamonas rubra* with 67% GC content on the other, and unsurprisingly, given their usually low GC content, separates the putative HGT regions from not only the *Pf*AR-1 whole genome but also from the vast majority of other *Pseudomonas* genomes. The second principal component also separates the putative HGT regions from the other genomes, although this component should not be over-interpreted because it accounts for only 6.5% of variation. The resulting plot loosely clusters the three GI regions with four sequenced *Pseudomonas* species, namely, *Pseudomonas luteola*, *Pinguicula lutea*, *Pseudomonas zeshuii* and *Pseudomonas* sp. HPB0071. Unfortunately, none of these four species were found to contain matches for the bait sequences in the cross-species comparison above, and thus, they are unlikely to be the origin of the putative HGT regions.

Regions which have been horizontally transferred have, by definition, an evolutionary history distinct from their host genomes. We, therefore, created a phylogenetic tree for the RE-like regions across the *Pseudomonas* clade, comprising the three RE regions from *Pf*AR-1, plus 72 RE-like regions from other species. This was then compared with a whole-genome phylogenetic tree of 280 *Pseudomonas spp.* supplemented by four more distant genomes, namely, *Azotobacter vinelandii* DJ, *A. baumannii* AC29, *E. coli* K12 MG1655, and *Burkholderia cenocepacia* J2315, which served as an outgroup. The 280 *Pseudomonas* genome subsets consisted of a) all 215 complete genomes, b) the 56 draft genomes showing a substantial hit against the Repeat Region bait set, as described above, and c) nine *Pseudomonas* genomes with unusual codon usage (*P. lutea*, *P. luteola*, *Pseudomonas* sp HPB0071, *Pseudomonas* sp FeS53a, *P. zeshuii*, *Pseudomonas hussainii* JCM, *P. hussainii* MB3, *Pedobacter Caeni*, and *Prauserella*

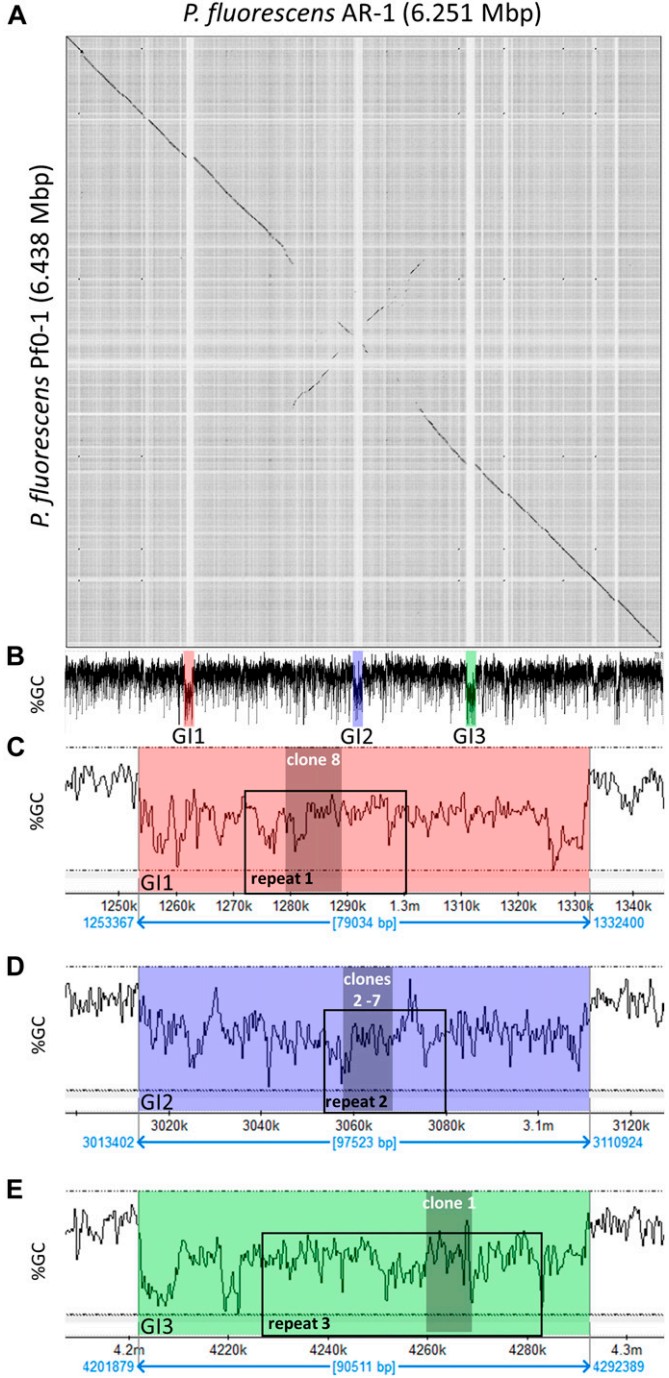

**Figure 3.  Genomic characteristics of *Pf*AR-1.**
**(A)** Dot plot alignment of the *Pf*AR-1 and *Pf*0-1 genomes. Numbering is from the putative origin of replication (*oriC*) loci. The disjunctions arising because of inserts in *Pf*AR-1 not present in *Pf*0-1 are clearly visible. **(B)** The GC content of the *Pf*AR-1 chromosome with GI1, GI2, and GI3 marked in red, blue, and green, respectively. **(C)** The low GC content region GI1 enlarged to show the position of repeat 1 (RE1) and the location of allicin resistance–conferring genomic clone 8. **(D)** The low GC content region GI2 enlarged to show the position of RE2 and the location of allicin resistance–conferring genomic clones 2–7. **(E)** The low GC content region GI3 enlarged to show the position of RE3 and the location of allicin resistance–conferring genomic clone 1.

*endophytica*). It is immediately apparent from comparison of the resulting region and whole-genome trees that the RE-like regions have a distinct evolutionary history (Supplemental Data 1). For independent confirmation of the above analysis, IslandViewer 4 (Bertelli et al, 2017) was used to assess the *Pf*AR-1 genome for HGT events. This analysis also clearly identifies the three putative HGT regions, although additional weaker candidate regions are also indicated (Fig S2).

## *ahpD*, *dsbA*, and *gor* can individually confer high allicin resistance

The contribution of individual genes to allicin resistance was investigated by transposon mutagenesis of clone 1 in *E. coli* and screening Tn mutants for loss of function. In addition, subcloning and overexpression of individual genes in *Ps*4612 was undertaken to assess for gain of function.

A decrease in allicin resistance compared with non-mutagenized genomic clone 1 was shown by 86 of 132 Tn mutants investigated in a streak assay. Tn mutants were examined by sequencing. No Tn insertions were found in the *osmC*, *sdr*, or *tetR* genes, but for most of the remaining genes, several independent Tn insertion sites were found. Tn mutants in each gene (Fig 4A) were tested for an increased allicin sensitivity phenotype in a drop test (Fig 4B). All Tn mutants grew less well in the absence of allicin stress than did controls (wt clone 1 and empty vector), as evidenced by the lower colony density visible at the $10^{-4}$ and $10^{-5}$ dilutions, respectively. No visible effect at either 150 or 200 µM allicin compared with clone 1 was observed for Tn insertions in the vector backbone, or the genes encoding the hypothetical protein or *kefF*, and the downstream region of *4-ot*. In contrast, Tn insertions in either *dsbA*, *trx*, *kefC*, *oye*, or *ahpD* led to a clear increase in allicin sensitivity at both concentrations. *ahpD*::Tn showed by far the highest sensitivity, and the phenotype resembled that of the empty vector control. *ahpD* potentially codes for an alkylhydroperoxidase, and the data suggest that this protein plays a major role in being able to confer allicin resistance to *Pf*AR-1. The contributions of the *dsbA* and *trx* genes to allicin resistance were more than those of the *kefC* and *oye* genes, but all of these Tn mutants showed a clear allicin sensitivity phenotype, especially at 200 µM allicin (Fig 4B).

The set of congruent genes on clone 1 were cloned individually in an expression vector to investigate the contribution of each gene to allicin resistance. *Ps*4612 was used for these experiments because we reasoned that even a small gain in resistance should easily be visible in this highly allicin-sensitive isolate. Only *ahpD* and *dsbA* conferred a gain of resistance when overexpressed individually. The resistance conferred by *ahpD* was almost as high as that conferred by the intact clone 1. Overexpression of *dsbA* in *Ps*4612 also caused a clear gain of resistance (Fig 4C).

Interestingly, both GI1 and GI2 have a Gor (glutathione reductase) gene (*gor2*, *gor3*) outside of the allicin resistance–conferring clones 2–8 in RE1 and RE2, respectively, and a further *gor* gene (*gor1*) is present on the *Pf*AR-1 chromosome. Because allicin oxidizes GSH to *S*-allylmercaptoglutathione, which is reduced by Gor to regenerate GSH (Horn et al, 2018), we investigated the potential contribution of Gor to *Pf*AR-1 allicin-resistance. Because *Pf*AR-1 has three *gor* genes, the experiments were performed with *E. coli*, which, like most bacteria, has only a single *gor* gene. Deleting the *gor* gene from *E. coli* BW25113 increased its sensitivity to allicin and resistance was

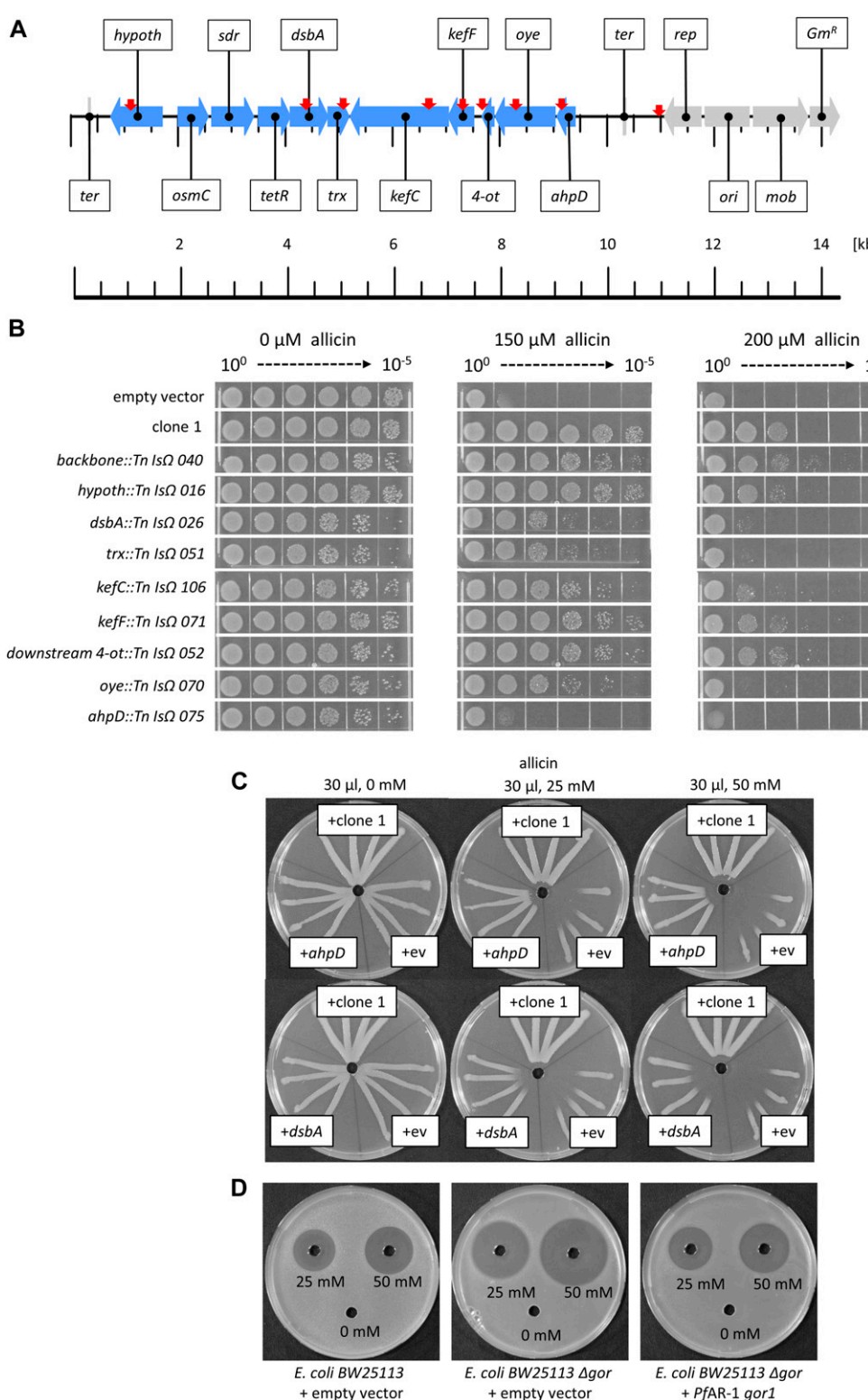

**Figure 4. Transposon mutagenesis of genes on clone 1.**
**(A)** Linear genetic map of *Pf*AR-1 genomic clone 1. *Pf*AR-1 genes are shown in blue, whereas genes on the vector backbone are shown in grey. The position of transposon insertions is indicated by red arrows. **(B)** *E. coli* MegaX DH10B transformed with clone 1, or empty vector, was compared with transposon insertion mutants in drop tests. All cultures were diluted to $OD_{600}$ = 1 (=$10^0$) and 5 $\mu$l of a $10^n$ dilution series down to $10^{-5}$ was dropped onto LB medium supplemented with different allicin concentrations. The experiment was performed twice. **(C)** Overexpression of *ahpD* or *dsbA* conferred allicin resistance to *Ps*4612. Test solutions were 30 $\mu$l, water, and 25 or 50 mM allicin. The experiment was performed twice. **(D)** *Pf*AR-1 glutathione reductase (*gor1*) complements *E. coli* BW25113 glutathione reductase deletion mutant (Δ*gor*). 40 $\mu$l of allicin solution (or water) were pipetted into wells in *E. coli*–seeded medium. The experiment was performed twice.

restored by complementing the Δ*gor* strain with chromosomal *Pf*AR-1 *gor1* (Fig 4D). These results clearly demonstrate the importance of Gor activity for allicin resistance, and in this connection, it is important to note that *Pf*AR-1 not only has three *gor* genes but also has a twofold higher basal Gor activity than *Pf*0-1 (Fig S3).

## Syntenic regions to *Pf*AR-1 REs in other plant-associated bacteria

Database searches revealed that some plant-associated bacteria, for example, the garlic pathogen *Pseudomonas salomoni* ICMP 14252 (Gardan et al, 2002) and a tomato- and *Arabidopsis thaliana* pathogen

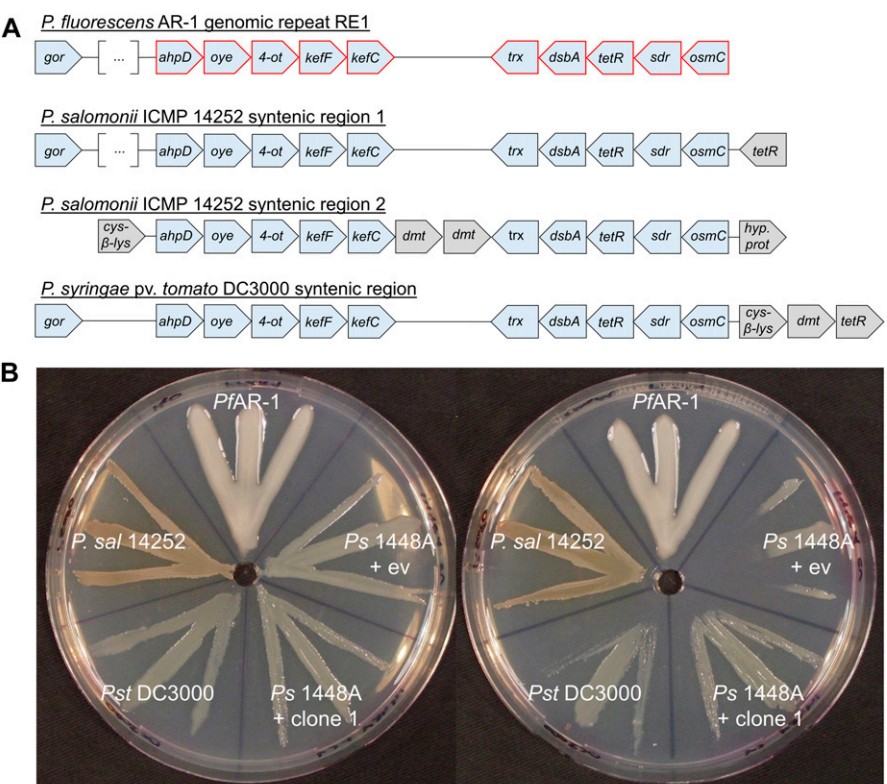

**Figure 5. The allicin resistance of *P. salomoni* ICMP 14252, *P. syringae* 1448A, and *P. syringae* DC3000 correlates with the number of syntenic regions that contain the core genes for allicin resistance identified in *Pf*AR-1.**

**(A)** A set of 10 genes is conserved in the genomic repeats of *Pf*AR-1 and in syntenic regions of *P. salomoni* ICMP 14252 and in *P. syringae* pv. *tomato* DC3000. *cys-β-lys*, cystathione-β-lyase; *dmt*, permease of the drug/metabolite transporter (*dmt*) superfamily; the remaining genes are referred to elsewhere in this study. The distance between the different genes does not represent the actual intergenic distances because the gene blocks were graphically aligned to highlight the conservation. In case of *gor* of *Pf*AR-1 RE1 and *P. salomoni* syntenic region 1, these genes are further upstream of the highlighted genes with several genes in between (represented by the squared brackets with three dots). Red highlighted genes represent the congruent set of genes also found in the resistance-conferring genomic clones of *Pf*AR-1. Coordinates of syntenic regions are: *P. salomoni* ICMP 14252 (GenBank: FNOX00000000.1) region 1 on contig 102 from position 324,974 to 392,566, and region 2 on contig 114 from position 73,863 to 86,381 and for *P. syringae* pv. *tomato* DC3000 (GenBank: NC_004578.1) from 4,794,584 to 4,807,117. **(B)** The allicin resistance of different bacteria correlates with the number of gene copies that are syntenic to the core fragment of the genomic clones from *Pf*AR-1. Ps1448A was either transformed with *Pf*AR-1 genomic clone 1 or pRU1097 (empty vector control), whereas the other strains were not genetically modified. *Pf*AR-1 has three copies of a set of 10 genes that were identified on genomic clones (e.g., genomic clone 1) that confer resistance to allicin in *P. syringae* strain 4612. *P. salomoni* ICMP 14252 has two copies of this set of genes in its genome, whereas *P. syringae* pv. *tomato* DC3000 has one and *P. syringae* 1448A none. The streak test was performed twice.

*P. syringae* pv. *tomato* DC3000 (*Pst* DC3000) (Buell et al, 2003) have regions syntenic with RE1. *Pf*AR-1 RE1 contains a *gor* gene and two gene groups (from *ahpD* to *kefC* and from *trx* to *osmC*) that are conserved in RE2 and RE3. These two groups are present in the two syntenic regions in the genome of *P. salomoni* ICMP 14252 and in one syntenic region in *Pst* DC3000 (Fig 5A). In contrast, the French bean (*Phaseolus vulgaris*) pathogen Ps1448A has no genes with significant similarity to any of the allicin resistance–conferring congruent gene set from *Pf*AR-1 clones. Ps1448A is fully sequenced (Joardar et al, 2005) and is quite similar at the nucleotide level to *Pst* DC3000 with an ANI of 86.87%. In comparison, the ANI between *Pf*AR-1 and *Pf*0-1 is 85.94%. A gene window analysis of *Pf*AR-1, *Pf*0-1, *Pst* DC3000, and *P. salomoni* ICMP 14252 suggests that the syntenic regions in DC3000 and ICMP 14252 were gained by HGT (Fig S4).

When *Pf*AR-1 (three copies), *P. salomoni* ICMP14252 (two copies), *Pst* DC3000 (one copy), and Ps1448A (no copies) were tested in a simple streak assay, we observed that *Pf*AR-1 and *P. salomoni* are most resistant against allicin, followed by *Pst* DC3000, then with a much higher sensitivity, by Ps1448A. The transfer of *Pf*AR-1 genomic clone 1 to Ps1448A raised its allicin resistance to approximately the same level observed in *Pst* DC3000 (Fig 5B).

## Discussion

The garlic defense substance allicin is a potent thiol reagent which targets the cellular redox buffer glutathione and accessible -SH groups in proteins (Borlinghaus et al, 2014). Allicin has been shown to *S*-thioallylate several cysteine-containing proteins in bacteria (Müller et al, 2016; Chi et al, 2019; Loi et al, 2019; Wüllner et al, 2019) and humans (Gruhlke et al, 2019) and has been described as a redox toxin (Gruhlke et al, 2010). *S*-thioallylation by allicin is reversible and sublethal doses suppress bacterial multiplication for a period of time, the length of which is dose-dependent, before growth resumes (Müller et al, 2016). Because allicin affects such a broad catalogue of cellular proteins, it is not easy for an organism to adapt to it by simple mutation. Thus, adding a lethal dose of allicin to a high-density bacterial culture and plating out for survivors, the routine strategy to isolate antibiotic-resistant mutants, has proven ineffective with allicin. Nevertheless, the sensitivity to allicin varies between different bacterial isolates, but the genetic basis for this variation is unknown. We reasoned that we would most likely find organisms with a high allicin resistance in association with the garlic bulb itself as a niche–habitat. This was indeed the case, and we were able to isolate the highly allicin-tolerant *P. fluorescens* Allicin Resistant-1 (*Pf*AR-1) from garlic. In inhibition zone tests, comparison with *E. coli* K12 DH5α or *P. syringae* 4612, *Pf*AR-1 showed an exceptionally high degree of allicin resistance (Fig 1A). To gain an insight into the mechanisms of allicin resistance in *Pf*AR-1, we used parallel approaches of functional testing of random genomic clones and whole-genome sequencing. Interestingly, genomic clones from *Pf*AR-1 were able to confer allicin resistance not only to closely related pseudomonads, but also to distantly related bacteria such as *E. coli* (Fig 1B and C).

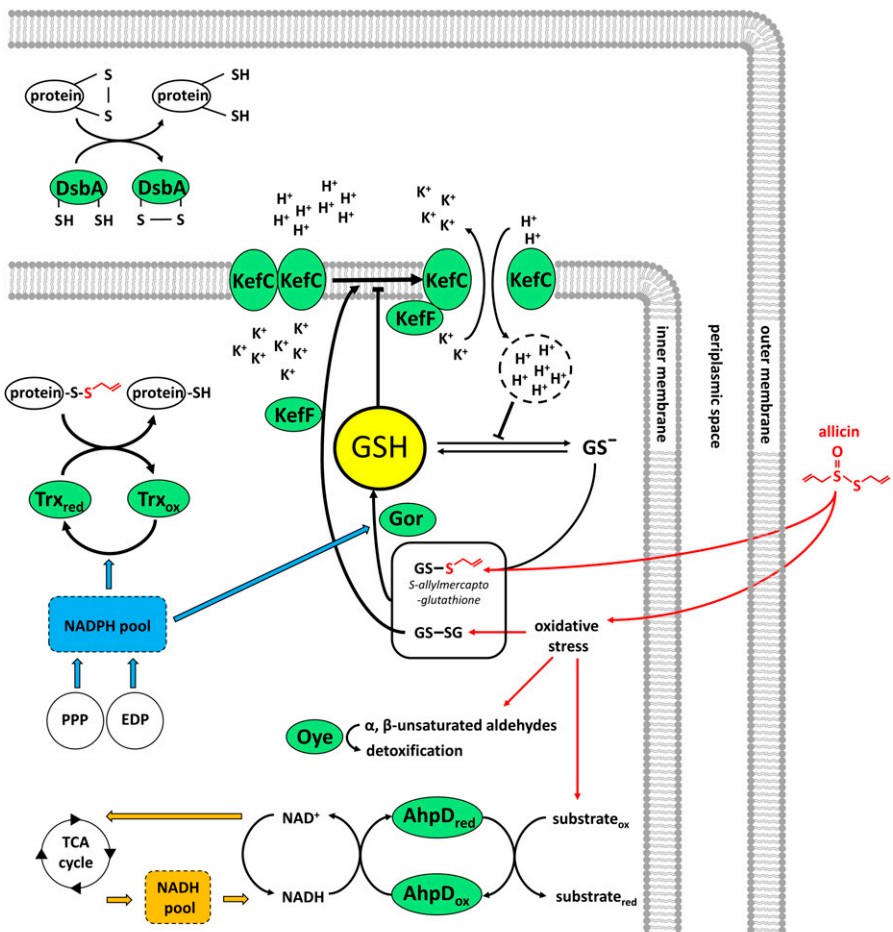

**Figure 6. Suggested model for allicin resistance in *Pf*AR-1.**
Green ovals show allicin resistance factors identified in *Pf*AR-1. AhpD$_{ox}$, AhpD$_{red}$, alkylhydroperoxidase D oxidized or reduced, respectively; DsbA, disulfide bond protein A; EDP, Entner–Doudoroff Pathway; Gor, glutathione reductase; GS$^−$, glutathione as the thiolate ion; GSH, glutathione; GS-SG, glutathione disulfide; KefF, KefC, glutathione-regulated potassium-efflux system; Oye, old yellow enzyme; PPP, pentose phosphate; protein-SH, protein with reduced cysteine; protein-S-SA, *S*-thioallylated protein; TCA cycle, Kreb's cycle; Trx, thioredoxin.

Resistance-conferring clones contained a congruent set of eight genes in common and 16 genes in total (Fig 2 and Table 1). Because allicin is a redox toxin causing oxidative- and disulfide stress, it was interesting to observe that half of these genes were annotated with redox-related functions (Fig 2B). Moreover, these genes were reported in the literature in the context of oxidative- and disulfide stress responses (Table 1).

Transposon mutagenesis of the resistance-conferring clones indicated that the *dsbA*, *trx*, *kefC*, and *oye* genes worked together, contributing incrementally to confer allicin resistance to a sensitive recipient. In contrast, the effect of a mutation in *ahpD* alone was major, with transposon mutants showing a similar phenotype to the sensitive parent transformed with empty vector (Fig 4B). These results are consistent with a multicomponent mechanism of allicin resistance. Annotated genes coding for significantly similar peptides were absent in the *Pf*0-1 reference strain, suggesting an external origin in *Pf*AR-1. This observation would explain why spontaneous mutation to gain of resistance upon allicin selection was not observed in axenic cultures of sensitive isolates under laboratory conditions.

The contribution of individual genes on the clones to the resistance phenotype was investigated by expressing them in highly susceptible *Ps*4612 cells. Expression of either *ahpD* or *dsbA* conferred a degree of allicin resistance almost as high as that conferred by the complete genomic clone (Fig 4C). In contrast, *trx* or *oye* expression had no obvious effect, although loss of function in transposon mutants caused an increase in allicin sensitivity (Fig 4B). This might indicate that the function of these genes depends on the function of another gene or genes from the genomic fragment, or that there are downstream effects of the Tn insertion. Overexpression lines for *osmC* and *kefC* were not recovered in *Ps*4612, most likely because of toxic effects. This may be due to the fact that the activity of KefC is normally tightly regulated by KefF and GSH, and an imbalance can lead to a toxic decrease in cellular pH and loss of potassium, which are important to maintain turgor and enable cell growth and division (Epstein, 2003).

*Pf*AR-1 genome analysis revealed unique features compared with the *Pf*0-1 reference strain. Three large genomic regions, between 79 and 98 kbp in size, with a lower GC content were identified (Δ%GC ~5–10%). These were designated GI1, GI2, and GI3 and they contained repeat regions RE1, RE2, and RE3, respectively, which encompassed the resistance-conferring clones (Fig 3B–E). Thus, the genome analysis and the functional studies independently identified the same set of genes. That these genes had been obtained by HGT was strongly indicated by codon usage analysis, which revealed differences in RE1, RE2, and RE3 compared with the core *Pf*AR-1 genome (Fig S1). Comparison with other *Pseudomonas* spp. suggested that the origin of the GIs was outside this genus. The HGT hypothesis was strongly supported by our phylogenetic analysis (Supplemental Data 1) and an independent in silico analysis using IslandViewer 4 (Fig S2). By current selection criteria regions RE1, RE2, and RE3, and most likely

the complete GI1, GI2, and GI3 regions, can reliably be considered to be bona fide GIs obtained by HGT. The preponderance of genes with redox-related functions in the REs fits well with the role in resistance against allicin. As previously noted, the presence of such large, widely spaced REs in the *Pf*AR-1 genome infers a high selection pressure to maintain them. Presumably, the latter relates to the allicin resistance–conferring function of the genes in question.

Although the GI-donor remains unknown, phylogenetic analysis identified similar syntenic regions to the REs from *Pf*AR-1 in several other pseudomonads (Fig 5A). Thus, the garlic pathogen *P. salomoni* ICMP14252 has two copies of the syntenic region, and the well-described model pathogen *P. syringae* pv. *tomato* DC3000 has one copy. The syntenic regions have the set of 10 core genes in clones 1–7 of *Pf*AR-1 (Fig 5A). Furthermore, the degree of allicin resistance correlates with the copy number. Isolates with multiple copies showed higher allicin resistance than those with only one or zero copies (Fig 5B). *P. salomoni* causes the café-au-lait disease on garlic (Gardan et al, 2002) and its high degree of allicin resistance corresponds well with its niche as a pathogen of garlic. One might expect that a pathogen like *P. salomoni* could be the origin of allicin resistance genes in *Pf*AR-1, but according to our codon usage analysis, the allicin resistance regions in *P. salomini* are quite distinct from the remainder of the genome and, therefore, were also likely obtained by HGT (Figs S1, S4, and Supplemental Data 1). *Pst* DC3000 is a model pathogen with a fully sequenced genome (Buell et al, 2003) that is pathogenic on tomato and on the model organism *A. thaliana* (Xin & He, 2013). To the best of our knowledge, the genes and their function in allicin resistance have not been described before in this well-studied strain. Our experiments suggest that the resistance conferred by the core region is specific to allicin-type oxidative stress and did not detectably increase resistance against other oxidants such as H$_2$O$_2$, CHPO, or NEM (Fig 1C). Nevertheless, oxidative stress has manifold causes, and some genes in the syntenic region may help to counter the manifold aspects of other forms of oxidative stress under some conditions. Thus, the oxidative burst in plants is a general defense response to avirulent pathogens (Lamb & Dixon, 1997). In this regard, it was reported that a transposon insertion in *dsbA* from the core genome of *Pst* DC3000 led to decreased virulence of *Pst* DC3000 on *A. thaliana* and on tomato (Kloek et al, 2000). Based on this study, it seems that the remaining *dsbA* copy from the syntenic region of *Pst* DC3000 was not sufficient to functionally complement the loss of the *dsbA* in the core genome, perhaps indicating a gene-dosage effect or subtly different functions between the two genes. It is intriguing to speculate that the syntenic region might help overcome the oxidative burst associated with plant defense, as well as protecting against more specifically redox-active sulfur-containing plant defense substances such as allicin, and it would be interesting to see if loss of syntenic genes other than *dsbA* in *Pst* DC3000 also leads to a reduction of virulence. Moreover, a recent study reported plasmid-born onion virulence regions in *Pantoea ananatis* strains that are pathogenic on onion (Stice et al, 2018, 2020 *Preprint*). The OVRA region contained a subset of genes that we describe in our present study as allicin resistance genes. More specifically, *dsbA*, which was annotated in *P. ananatis* OVRA as "isomerase," *oye* (as *alkene reductase*), *trx*, *ahpD* (annotated as *alkylhydroperoxidase*), *glutathione disulfide reductase*, *sdr*, and *osmC*, were all present. Although onion does not produce allicin,

upon damage, it accumulates small amounts of other thiosulfinates and other sulfur-containing redox-active compounds which may be involved in defense (Block et al, 1992; Lawson et al, 1991b; Imai et al, 2002; Block, 2010). Nevertheless, there are several plant-pathogenic bacteria, for example, the bean pathogen *P. syringae* 1448A, which have no equivalent syntenic region but are successful plant pathogens in their own right. Therefore, there is clearly no absolute requirement for the syntenic region to enable colonization of plants as a habitat per se. In this regard, it should be noted that a comprehensive genomic analysis of plant-associated bacteria to identify protein domains associated with adaptation to growth in or on plants showed that seven of the 10 genes we identified in the syntenic region contained plant-associated domains as described by the authors (Levy et al, 2018). A list of pseudomonads with syntenic regions similar to those in *Pf*AR-1 is shown in Table S2.

Allicin targets inter alia the GSH pool in plants, and GSH metabolism has been shown to be important in the resistance of bacteria, yeast, and *A. thaliana* to allicin (Gruhlke et al, 2010; Müller et al, 2016; Leontiev et al, 2018). In the work reported here, we show that *Pf*AR-1 has three copies of the glutathione reductase (*gor*) gene, one copy each on RE1 and RE2, but outside the core region represented in clones 1–8, and one copy in the core genome. This is quite remarkable because bacteria normally have only one *gor* gene. Exceptions, such as *Pst* DC3000 and *P. salomoni* ICMP14252, have an additional *gor* gene that was also very likely obtained by HGT as in *Pf*AR-1. We demonstrated that the high *gor* copy number in *Pf*AR-1 correlated with a twofold higher basal Gor enzyme activity compared with *Pf*0-1 with only one copy of *gor* (Fig S3). The importance of Gor activity for tolerance to allicin was shown by the enhanced sensitivity of an *E. coli* Δ*gor* knockout and the complementation of this phenotype by *gor1* from *Pf*AR-1 (Fig 4D). Gor recycles oxidized glutathione (GSSG) to GSH. GSH protects cells from oxidative stress, either by direct reaction with pro-oxidants such as allicin, thus scavenging their oxidative capacity (Fig 6), or by serving as an electron donor for detoxifying enzymes such as glutathione peroxidase and glutaredoxins (Meister & Anderson, 1983). It was shown that allicin treatment leads to oxidation of GSH to GSSG in yeast (Gruhlke et al, 2010) and to the formation of *S*-allylmercapto-glutathione (GSSA) (Horn et al, 2018). In yeast, both GSSA and GSSG are reduced by Gor to release GSH (Horn et al, 2018). Gram-positive bacteria, such as *Staphylococcus aureus*, have bacillothiol rather than GSH and in an independent investigation, we showed that the bacillothiol reductase YpdA, which is the functional equivalent of Gor, reduced *S*-allylated bacillothiol (BSSA). YpdA was important for the resistance of *Staphylococcus* to allicin (Loi et al, 2019). Furthermore, GSH negatively regulates the activity of KefC, but GSH conjugates stimulate KefC activity via KefF (Ferguson et al, 1997; Miller et al, 2000) (Fig 6). Thus, GSH inhibits K$^+$ efflux and *E. coli* Δ*gsh* mutants lose K$^+$ ions similarly to cells stressed with electrophiles such as NEM (Meury & Kepes, 1982; Elmore et al, 1990). KefC activity acidifies the cytoplasm and has been reported to protect against oxidative stress caused by electrophiles such as NEM and methylglyoxal, presumably because the lowered pH works against thiolate ion formation (Ferguson et al, 1993, 1995, 1996, 1997; Poole, 2015). KefC activation could be expected to protect against oxidative stress caused by the electrophile allicin in the same way (Fig 6). Thus, some of the genes in the core fragment might be expected to help bacteria to be less sensitive to other oxidants. However, this effect was apparently not major enough to be

observed in the tests documented in Fig 1C, where only a reduced sensitivity to allicin-type stress was clearly observed.

Gor uses NADPH as a reductant and the pentose phosphate pathway (PPP) is the major source for NADPH in most cells. It has been shown that yeast mutants compromised in the NADPH-producing steps of the PPP are hypersensitive to allicin (Leontiev et al, 2018). Because PfAR-1 lacks the 6-phospho-fructokinase gene necessary for glycolysis, it depends on the Entner–Doudoroff Pathway (EDP) to metabolize glucose to pyruvate, and this yields NADPH in addition to NADH. Thus, the EDP confers an advantage during oxidative stress by providing an additional source of NADPH for Gor in addition to that from the PPP (Conway, 1992). It was shown for Pseudomonas putida that key enzymes of the EDP are up-regulated upon oxidative stress (Kim et al, 2008). NADPH is also used as reducing equivalents by antioxidative enzymes such as Oye-dehydrogenases. Moreover, in Mycobacterium tuberculosis, the AhpD enzyme depends on NADH consumption (Bryk et al, 2002), and thus, PfAR-1 could be able to tap into two pools of reducing equivalents, both NADPH and NADH, to defend against allicin stress (Fig 6).

Disulfide bond protein A (DsbA) is located in the periplasm in E. coli (Shouldice et al, 2011), and based on its protein domain content, in PfAR-1 DsbA might act as disulfide isomerase or as a chaperone (Fig 6). In E. coli, a part of the Dsb system is supported via thioredoxin-reducing equivalents from the cytosol (Trx) (Katzen & Beckwith, 2000). The extra Trx copies in PfAR-1 might be important in this regard during allicin stress. How alkylhydroperoxidase D (AhpD) might protect PfAR-1 against allicin stress to such a high degree is so far unclear. Possibly, as in M. tuberculosis, it might act by using NADH to reduce oxidized molecules arising from oxidative stress (Bryk et al, 2002) caused by allicin (Fig 6).

Taken together, our data reinforce the central importance of GSH metabolism and redox enzymes in the resistance of cells to the electrophilic thiol reagent allicin and identify specific genes important for the multicomponent resistance mechanism. The maintenance of multiple copies of resistance genes, obtained by HGT, probably facilitates exploitation of the garlic ecological niche by PfAR-1 in competition with other bacteria.

# Materials and Methods

Additional information about bacterial strains, plasmids, and primers are given in Tables S3–S5.

## Cultivation methods and media

E. coli was routinely cultivated at 37°C in 2×YT medium (Sambrook & Russel, 2001).

Pseudomonads were routinely cultivated at 28°C in King's B medium (King et al, 1954). In contrast to the original recipe, $MgSO_4$ was left out of the King's B medium in this study.

M9JB medium was developed during this study for the cultivation of Pseudomonas for reduced slime production. This defined medium is based on M9 salts (Maniatis, 1982) with glycerol as carbon source (1.25% wt/vol). In addition, 1× Nitsch vitamin mixture (product N0410; Duchefa Biochemie) was added to complement for E. coli auxotrophies and

3× complete supplement mix (product DCS0019; Formedium) to enrich the media for amino acids (except cysteine) to improve doubling time.

## Inhibition zone assays

Bacteria were freshly grown from an optical density at 600 nm ($OD_{600}$) of 0.05 to $OD_{600}$ = 0.2 – 0.3. Bacteria-seeded agar was prepared by dispersing 300 $\mu$l of liquid culture in 20 ml 50°C warm agar medium and pouring immediately into Petri dishes. A surface lawn of bacteria was prepared by spreading 125 $\mu$l of an $OD_{600}$ = 1.0 culture onto 20 ml of solidified agar in a Petri plate. Bacteria were spread over the surface with glass beads ($\varnothing$ = 3 mm) by gentle shaking. Wells ($\varnothing$ = 0.6 cm) were punched out of the solidified agar with a cork borer to apply the test solution. Plates were then incubated overnight.

## Streak tests

A single bacterial colony was picked and suspended in the liquid medium, then streaked away from a central well ($\varnothing$ = 0.6 cm) in 20 ml of solid medium in a Petri plate. Test solutions were pipetted into the central well.

## Drop tests

Overnight E. coli suspension cultures were adjusted to $OD_{600}$ = 1.0 and $10^n$ dilution series to $OD_{600}$ = $10^{-5}$ were prepared. Aliquots (5 $\mu$l) of each dilution were dropped on solid media (2×YT) containing different amounts of allicin. Plates were incubated at 37°C overnight.

## Chemical synthesis of allicin

Chemical synthesis of allicin was performed as described previously, with the exception that the allicin was not dried with $MgSO_4$ but directly dissolved in $H_2O$ and used without further column purification. Purity and quantity was checked via HPLC analysis (Gruhlke et al, 2010).

## Protocol for high-yield genomic DNA (gDNA) extraction from bacteria

For preparing a gDNA library of PfAR-1, a protocol for high-yield DNA extraction was established based on Chen and Kuo (1993) and on Syn and Swarup (2000). A 50-ml bacterial culture was grown overnight in liquid medium in a 500-ml Erlenmeyer flask. Bacteria were harvested by centrifugation (2,500g for 20 min at 4°C) in a 50-ml reaction tube. The cell pellet was suspended in 20 ml of 1% NaCl solution (wt/vol in double-distilled water [$H_2O_{dd}$]) for the removal of bacterial exopolysaccharides. Therefore, the cells were vortexed vigorously in the NaCl solution and harvested again by centrifugation. For removal of NaCl, the bacterial cells were washed twice with 50 ml $H_2O_{dd}$ by vigorous vortexing and harvesting by centrifugation. The cells were finally suspended in 40 ml $H_2O_{dd}$. The cell solution was distributed among 2 ml reaction tubes and harvested at 12,879g for 3 min at 4°C. Afterwards, the supernatant was removed to the last drop. The cell pellets were vortexed without addition of buffer to loosen the cells from each other, thereby increasing the available surface for the subsequent lysis step. Bacterial lysis was performed by addition of 1.36 ml lysis buffer (40 mM TRIS–HCl, pH 7.8, 20 mM sodium acetate, 1 mM EDTA, and

1% SDS [wt/vol = 35 mM]) to each reaction tube and mixing by pipetting up and down. The tubes were then incubated for 60 min in a 50°C water bath for enhanced lysis and DNA yield. Then, 12 $\mu$l of RNase I (10 mg/ml) were added to each reaction tube and incubated for 30 min at 37°C. To precipitate cell debris and SDS, 476 $\mu$l 5 M NaCl were added to each reaction tube and mixed gently. The cell debris and SDS were then separated from the remaining solution via centrifugation at 20,937$g$ for 20 min at 4°C.

For further purification, 1.6 ml from the supernatant of each reaction tube was gathered in an autoclaved glass bottle. Afterwards, the bottle was filled up with dilution buffer (40 mM TRIS–HCl, pH 7.8, 20 mM sodium acetate, 1 mM EDTA, and 150 mM NaCl) to ~200 ml for dilution. The bottle was placed on ice.

For phase extraction, 5 ml of chloroform were added to 40 ml centrifugation tubes. The tubes were then filled up with the DNA solution which were gathered previously in the glass bottle and inverted 50 times. The phases were separated by centrifugation at 21,000$g$ for 3 min at 4°C. The supernatant was gathered in a new sterile glass bottle. These extraction steps were repeated for the whole DNA solution in the glass bottle until no interphase was visible any more.

For DNA precipitation, 25 ml of phase-extracted DNA solution was added to 50-ml reaction tubes and mixed with 25 ml isopropanol. Because the lysis buffer and the dilution buffer contained enough salt (not removed during former steps), no further salt addition was needed for precipitation. The DNA-isopropanol/solutions were stored at –20°C until all the remaining solution was processed to this stage of this protocol.

The DNA was subsequently precipitated into the same tubes at 21,000$g$ for 15 min at 4°C. The two DNA pellets were washed twice with 70% ethanol. Last droplets of ethanol were removed via a Pasteur pipette. DNA pellets were dissolved in 10 mM of TRIS–HCl, pH 8. The DNA was then aliquoted and stored at –20°C.

### *Pf*AR-1 genomic library construction

gDNA was extracted as described and partially digested with Sau3AI FD (Thermo Fisher Scientific). Sau3AI FD was diluted 300-fold in 1× FastDigest buffer (Thermo Fisher Scientific) and was applied to the reaction mixture for a 3,000-fold enzyme dilution. Digested DNA was size-separated via agarose gel electrophoresis and fragments of ~10 kbp were extracted and purified using Zymoclean Large Fragment DNA Recovery Kit, subcloned in BamHI-digested pRU1097, and electroporated in *E. coli* K12 DH10B MegaX. Plasmid DNA of ~14,000 *E. coli* transformant colonies was extracted, representing more than 99.99% theoretical coverage of the *Pf*AR-1 genome.

### Transposon mutagenesis of genomic clone 1

Transposon mutagenesis of *Pf*AR-1 genomic clones on pRU1097 was performed in the *Ps*4612 background. Briefly, pSCR001 carrying transposon IS-Ω-km/hah was transferred from *E. coli* S17 via biparental mating to *Ps*4612, and transconjugants were selected on gentamycin and kanamycin. Because pSCR001 cannot replicate in *Ps*4612, plasmid isolation from the transconjugants yields a Tn-carrying pRU1097 population, which was transformed in *E. coli* MegaX DH10B. Plasmid DNA of more than 10,000 *Ps*4612 genomic clone 1 transconjugants was extracted and electroporated in *E. coli* K12 DH10B MegaX to construct a library of Tn-carrying genomic clone 1.

### PCR applications and DNA cloning

All DNA cloning steps in this work were based on enzymatic restriction and sticky end (or blunt end) DNA ligation with T4-DNA ligase from Thermo Fisher Scientific, except the construction of pJABO5 and the subsequent cloning of *gor* genes (see below). The necessary restriction sites for PCR fragments were introduced during PCR via primer overhangs if not already present in the DNA template.

For all PCR applications, the Phusion High-Fidelity PCR Master Mix (Thermo Fisher Scientific) was used according to the user manual.

### Construction of the broad host range expression vector pJABO

Linearized pRU1097 was amplified via PCR with primers P163 and P174, thus adding ApaLI and XhoI restriction sites at the ends. The promoter from the neomycin phosphotransferase gene from pJP2neo was amplified with the primers P160 and P159, thus adding the restriction sites NheI, PvuI, and XhoI upstream and ApaLI downstream of the promoter, respectively. Both the above PCR products were digested with ApaLI and XhoI and ligated together to give the pRU1097+Neo promoter intermediate.

Next, the multiple cloning site (MCS) from pBluescript I KS (-) was amplified with the primers P161 and P162, thus adding the restriction sites NheI and PvuI After restriction with PvuI and NheI, this was ligated with pRU1097+Neo promoter to give pRU1097+Neo+MCS.

The NotI restriction site in the mobilization gene (Mob) from pRU1097+Neo+MCS was removed by whole vector amplification using the primers P183 and P184 and subsequent blunt-end ligation. Primer P183 introduces a nucleotide exchange within the recognition sequence for NotI, resulting in the deletion of NotI without changing the encoded amino acid. The elimination of the restriction site was checked via restriction analysis and the constructed vector was analyzed by DNA sequencing. Sequencing showed that all components for gene expression except for the *rrnB1* terminator sequence were present.

To restore the *rrnB1* terminator somehow lost during the previous steps, the sequence was reamplified from pRU1097 with the primers P217 and P220, adding SacI and PvuI restriction sites for subcloning.

The final vector construct pJABO was verified by restriction analysis and DNA sequencing of the promoter and the MCS as well as their flanking terminator sequences *T4* and *rrnB1*.

### Construction of the broad host range vector pJABO5 and cloning of *Pf*AR-1 glutathione reductase *gor1* gene for inducible expression in *E. coli*

pJABO5, which was used for the expression of the *Pf*AR-1 glutathione reductase in *E. coli*, was constructed by in vivo recombination in yeast. In comparison with pJABO, which was used for overexpression, pJABO5 was designed for induced gene expression based on the inducible lac promoter from *E. coli*.

pRU1097 was digested overnight with XbaI and SacI, thereby removing *GFP* from pRU1097. Next, *yeast* 2 μ *ori* and the *URA3* selection marker were amplified from pRS426 via PCR using the primers P449 and P506. The *lac* promoter was amplified from *E. coli* MG1655 gDNA with primers P488 and P507, and the *lacZ* fragment was amplified from pBluescript I KS (-) using the primers P489 and P490. The vector

backbone fragment of pRU1097 and the PCR products were transformed in *Saccharomyces cerevisiae* BY4742 as described in Jansen et al (2005). The vector was extracted from yeast by alkaline lysis and retransformed into *E. coli* for amplification.

For cloning of *Pf*AR-1 *glutathione reductase 1, gor1* had to be amplified via a nested PCR because the different glutathione reductases within the core genome and the horizontally transferred regions were too similar for separate, one-step amplification. Thus, the first PCR amplicon from *Pf*AR-1 gDNA was generated with the primers P323 and P324 and used as a template for the amplification of *Pf*AR-1 *gor1* with the primers P524 and P525. The final product was cloned in pJABO5 by in vivo recombination in yeast (Jansen et al, 2005). pJABO5 was digested with BamHI and Lac*Zα* was replaced by *Pf*AR-1 *gor1*. The recombinant vector was isolated from yeast and directly transformed in *E. coli* BW25113 wild type or *E. coli* BW25113 Δ*gor*. The presence of the subcloned *gor1* was verified by PCR using the primers P195 and P491.

### Protein extraction and glutathione reductase activity assay

Pseudomonads were grown overnight in liquid M9JB medium to decrease slime production. Crude bacterial cell lysate was prepared from bacteria by vortexing with glass beads. Glutathione reductase activity assay was performed as described.

### Glutathione disulfide reductase enzyme assay

For glutathione reductase activity assays, the cells were grown overnight at 28°C in liquid M9JB medium. Cells from 20 ml overnight culture were harvested by centrifugation (3,000*g* at room temperature) and they were resuspended in 1 ml phosphate buffer (143 mM Na-phosphate containing 6.2 mM EDTA, pH 7.5). Bacteria were lysed mechanically by vortexing with 1-mm glass beads three times for 1 min on ice. Cell debris were removed by centrifugation at 21,000*g* for 1 min at room temperature.

Glutathione reductase activity was measured in a glutathione reductase recycling assay (Horn et al, 2018) modified to conditions showing linear dependency of the reaction velocity for enzyme amount, that is, not substrate-limited. Absorption was followed over 10 min at 412 nm using a spectrophotometer (DU800; Beckman Coulter GmbH). Enzyme activity was calculated assuming a molar extinction coefficient of TNB of 13,600 $M^{-1} \cdot cm^{-1}$ (Ellman, 1959). For calculation of specific enzyme activity, protein content of the sample was measured using the Bradford method (Bradford, 1976).

### Genome sequencing of *Pf*AR-1

*Pf*AR-1 was grown in KB medium in a rotary shaker at 200 rpm and 28°C overnight. For DNA extraction, 15 ml of overnight culture was washed three times in 1×TE with 50 mM EDTA by repeated pelleting at 5,000*g* and resuspension by vortexing. The subsequent cell lysis was performed as described by Sambrook and Russel (2001) for Gram-negative bacteria. From this material, three Illumina paired-end libraries were created and run multiplexed in conjunction with other samples, twice as 2 × 100 paired-end runs on a HiSeq 2000, and once as a 2 × 311-bp paired-end run on a MiSeq. The resulting data were filtered by Trimmomatic V0.32 (Bolger et al, 2014) and assembled using SPAdes V3.5.0 (Bankevich et al, 2012). The resulting

assembly was largely complete, with a total size of 6.3 Mbp, but it was still relatively fragmented with 40 scaffolds of 1 kbp or larger and an N50 of 370 kbp.

To fully resolve the genome into one contig, two additional long read datasets were generated on the Pacific Biosciences RS-II platform. For DNA extraction, 15 ml of overnight culture were washed three times in 1×TE with 50 mM EDTA by repeated pelleting at 5,000*g* and resuspension by vortexing. The subsequent cell lysis was performed as described in Sambrook and Russel (2001) for Gram-negative bacteria. Further depletion of contaminating polysaccharides was achieved by application of the Pacific Biosciences protocol (Pacific Biosciences, 2019) for gDNA cleanup. The final DNA was eluted in RNase-free water and quality was determined using NanoDrop for purity and Qubit for quantification. Sequencing was performed by GATC Biotech AG. The resulting two datasets, combined with the Illumina datasets described above, were then assembled, using SPAdes 3.5.0, yielding a single contig sequence of ~6.26 Mbp.

Self-alignment of this contig revealed that 9,642-bp sequence was duplicated on each end which was then removed from one end. To simplify cross-genome comparisons, this sequence was aligned against the *Pf*0-1 reference sequence, and oriented to match, resulting in the 6,251,798-bp *Pf*AR-1 assembly. The completed genome was then submitted to the RAST webserver (Aziz et al, 2008; Overbeek et al, 2014; Brettin et al, 2015) for automatic structural and functional annotation.

### In silico analysis of the *Pf*AR-1 genome

The low-GC regions identified in the *Pf*AR-1 genome were initially compared manually by cross-referencing the functional annotation of genes. This revealed a list of genes from each region which have a potentially common origin. After removing low-confidence protein annotations, which were both unique to a single region and lacking a definitive functional annotation, namely, *two hypothetical proteins*, the remaining genes were manually reconciled into a putative ancestral arrangement of 26 genes.

### Comparison of putative HGT regions across the *Pseudomonas* genus

A set of bait genes was created based on the putative 26-gene ancestral arrangement described above. Because these 26 groups were generally represented in more than one region, the set comprised 57 sequences in total. All available *Pseudomonas* sequences, comprising 215 complete genomes and 3,132 draft genomes, were downloaded from the Pseudomonas Genome Database Web site (https://www.pseudomonas.com/) and queried for the bait sequences using BLAST. Similarity was calculated using a sliding window of 40 genes, and regions which exceeded a normalized bit-score total of five were selected.

### Interspecies codon analysis

Synonymous codon usage statistics were calculated for the full *Pf*AR-1 genome, the three putative HGT regions, the 3,347 other available *Pseudomonas* genomes, and eight representative non-*Pseudomonas* Gammaproteobacteria (*A. baumannii* AC29, *Alkanindiges illinoisensis*, *A. vinelandii* DJ, *E. coli* K12 MG1655, *Moraxella catarrhalis*, *Perlucidibaca piscinae*, *R. rubra*, and *Ventosimonas gracilis*). After removing methionine

and tryptophan, which have only one codon, the remaining codons were analyzed using principal component analysis.

### Gene window codon analysis of *Pf*AR-1, *Pf*0-1, *Pst* DC3000, and *P. salomoni* ICMP 14252

From the 3,347 publicly available genomes, three were selected, in addition to *Pf*AR-1, for assessment of local codon usage using a 10-gene sliding window approach. These three genomes were *Pf*0-1, as the reference *Pseudomonas* strain closely related to *Pf*AR-1, although lacking any putative HGT region; *Pst* DC3000, a well-studied plant pathogen, which contained one putative HGT region; and *P. salomoni* ICMP14252, a garlic pathogen which contains two putative HGT regions.

### Phylogenetic comparison of whole genome versus RE-like sequences

Whole-genome phylogenetic analysis was performed using Ortho-Finder (Emms & Kelly, 2015; version 1.1.8, https://github.com/davidemms/OrthoFinder/releases/tag/1.1.8) to place the newly sequenced *Pf*AR-1 genome in its phylogenetic context, using a subset of 280 *Pseudomonas* genomes supplemented by four more distant genomes downloaded from National Center for Biotechnology Information GenBank, namely, *A. vinelandii* DJ, *A. baumannii* AC29, *E. coli* K12 MG1655, and *B. cenocepacia* J2315 which served as an outgroup. The 280 *Pseudomonas* genome subset consisted of a) all 215 complete genomes, b) the draft genomes showing a substantial hit against the putative-HGT gene set, as described above, and c) nine *Pseudomonas* genomes with unusual codon usage (*P. lutea*, *P. luteola*, P. sp HPB0071, *P.* sp FeS53a, *P. zeshuii*, *P. hussainii* JCM, *P. hussainii* MB3, *P. caeni*, and *P. endophytica*).

In a second analysis, the three putative-HGT from *Pf*AR-1 were compared against the corresponding regions from other *Pseudomonas* genomes, identified as described above. For this analysis, the sequences from each GI region were re-ordered according to the best match against the 26 bait group sequences, concatenated to form a single pseudo-sequence and aligned using MAFFT (version 7, [Katoh & Standley, 2013]). The resulting multiple alignment was accessed using "fitch" from Phylip (version 3.69) and the resulting trees were visualized using FigTree (version 1.4.3, https://github.com/rambaut/figtree/releases/tag/v1.4.3).

### IslandViewer analysis

For independent confirmation of the HGT analysis, the *Pf*AR-1 genome was submitted to the IslandViewer 4 (Bertelli et al, 2017) Web site, for assessment regarding HGT events.

### Additional annotation of genomic repeat regions

Gaps in the annotation of genomic repeats with putative horizontal origin indicated incomplete annotation, also implicated by a low gene density (1 gene per 1.3–1.6 kbp), which is expected to be one gene per 1 kbp in bacterial genomes (Koonin & Wolf, 2008). Regions were submitted individually without the remaining genome sequence to the RAST webserver, thereby closing annotational gaps (1 gene per 0.90 kbp in average). Remaining DNA regions without annotation were manually curated using National Center for Biotechnology Information open reading frame finder and BLASTp.

### Dot plot and %GC content analysis

For dot plot analysis and %GC content analysis and comparison, Genome Pair Rapid Dotter (GEPARD, [Krumsiek et al, 2007]), Artemis Comparison Tool (Carver et al, 2005), and UGENE (Okonechnikov et al, 2012) were used, respectively.

### Congruent set of genes and copy number analysis

Analysis was performed by batch translation of the coding sequences of the *Pf*AR-1 genomic repeats into peptide sequences using coderet from the emboss suite (Rice et al, 2000) and compared these against all other peptide sequences from the genomic repeats and the remaining genome, respectively. Peptides with a minimal peptide length of ≥100 amino acids were compared using BLASTp combined with the graphical user interface visual blast (Mele, 2016). Significantly similar sequences were defined by a minimal sequence similarity of ≥25% and with an E-value ≤ 0.0001.

## Data Availability

The *Pf*AR-1 genome sequence is available at European Molecular Biology Laboratory - European Bioinformatics Institute under project PRJEB34663.

## Supplementary Information

## Acknowledgements

Nikolaus Schlaich and Jürgen Prell are thanked for helpful discussions and Ulrike Noll for proof-reading the manuscript. Financial support from the Rheinisch-Westfälische Technische Hochschule Aachen University (J Borlinghaus, AJ Slusarenko, MCH Gruhlke) is gratefully acknowledged. J Borlinghaus was supported by an RFwN PhD stipendium and A Bolger by Bundesministerium für Bildung und Forschung (BMBF) grant 031A536. This research did not receive any specific grant from funding agencies in the public, commercial, or not-for-profit sectors.

### Author Contributions

J Borlinghaus: conceptualization, investigation, visualization, methodology, and writing—original draft, review, and editing.
A Bolger: conceptualization, data curation, software, formal analysis, investigation, methodology, and writing—original draft, review, and editing.
C Schier: investigation and methodology.
A Vogel: investigation and methodology.
B Usadel: writing—review and editing.
MCH Gruhlke: methodology and writing—original draft, review, and editing.

AJ Slusarenko: conceptualization, resources, supervision, visualization, project administration, and writing—original draft, review, and editing.

## Conflict of Interest Statement

The authors declare that they have no conflict of interest.

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
