## [Reviewer comments · Life Science Alliance]

Life Science Alliance

Genetic and molecular characterization of multi-component resistance of *Pseudomonas* against allicin

Jan Borlinghaus, Anthony Bolger, Christina Schier, Alexander Vogel, Björn Usadel, Martin Gruhlke, and Alan Slusarenko

DOI: <https://doi.org/N/A>

Corresponding author(s): Alan Slusarenko, RWTH Aachen University and Jan Borlinghaus, RWTH Aachen University, Department of Plant Physiology

Review Timeline:	Submission Date:	2020-02-07
	Editorial Decision:	2020-02-21
	Revision Received:	2020-03-14
	Accepted:	2020-03-16

Scientific Editor: Andrea Leibfried

Transaction Report:

February 21, 2020

RE: Life Science Alliance Manuscript #LSA-2020-00670-T

Alan John Slusarenko
RWTH Aachen University
Plant Physiology (Bio3)
Worringerweg 1
Aachen 52056
Germany

Dear Dr. Slusarenko,

Thank you for submitting your manuscript entitled "Genetic and molecular characterization of a multi-component resistance mechanism of a pseudomonad against allicin". Your work has now been peer-reviewed by two experts and I attach their reports below.

As you will see, the reviewers appreciate your data and only have minor suggestions for improvement. We would thus like to invite you to submit a revised version of your manuscript to us. While responding to the specific concerns of the two reviewer, please also pay attention to the following:

- Please make sure to fill in all mandatory fields within our submission system when uploading the revised version
- Please provide the final version of your manuscript in docx format
- Please upload all figure files, including supplementary figures, as individual files and without figure legends, the legends (main figures and S figures) should remain in the main manuscript docx file
- Please incorporate the supplementary information within the manuscript to allow others to more easily appreciate it
- I couldn't find the deposited genome data - please provide information on the exact database and accession code

Please get in touch in case you would like to discuss individual revision points further. Please also provide a point-by-point response when submitting the final version of your paper.

A. FINAL FILES:

B. MANUSCRIPT ORGANIZATION AND FORMATTING:

Thank you for this interesting contribution, we look forward to publishing the revised version of your paper in Life Science Alliance.

Sincerely,

Andrea Leibfried, PhD
Executive Editor
Life Science Alliance
Meyerhofstr. 1
69117 Heidelberg, Germany

t +49 6221 8891 502
e a.leibfried@life-science-alliance.org
www.life-science-alliance.org

Reviewer #1 (Comments to the Authors (Required)):

This manuscript provides genetic and molecular characterization of the allicin resistance mechanism of the garlic-adapted bacterium *Pseudomonas fluorescens*. There is a high degree of novelty in this well-written, experimentally solid manuscript, which with minor changes merits publication as is.

Minor errors:

- 1) mis-use of semicolons where commas are needed (line 50 after close-parenthesis and line 490)
- 2) The statement on lines 450-453 is incorrect "Although onion does not produce allicin, it has many sulfur containing redox-active compounds which may be involved in defense (Imai et al. 2002), for example diallyl tetrasulfane, which was recently shown to S-thioallylate a similar spectrum of proteins to allicin in *Bacillus subtilis* (Chi et al. 2019)." While onion does contain various sulfur compounds, there are no allyl compounds present so "diallyl tetrasulfane" cannot be correct.
- 3) The referencing style is inconsistent with some titles containing all first letters capitalized and some not. Shouldn't all botanical terms in titles be italicized as is done in the manuscript text?

Apart from these minor errors, this is an excellent manuscript meriting publication without change.

Reviewer #2 (Comments to the Authors (Required)):

The manuscript by Borlinghaus et al. describes a comprehensive study of a recently discovered pseudomonad strain resistant to the highly aggressive redox modulator allicin. The authors demonstrate that this resistance is most likely acquired by horizontal gene transfer and results from the interplay of several proteins and enzymes, of which about half are redox related. Overall this is a very interesting manuscript, nicely written and presented, and of considerable significance for the field of redox biology.

Some parts, such as the section on pages 10 and 11, are very exhaustive. Then again, I am not an expert in gene transfer and such details may be of interest to colleagues active in this field.

I have spotted a few very minor mistakes which may be corrected as part of a minor revision. These are:

line 48: replace protein with proteins

line 78: replace work with study

line 384: replace lab with laboratory

line 444: replace lead with leads

line 452: check if correct as onions tend to contain the propyl- rather than allyl-derivatives

line 453 and 457, and thereafter: Repetitive employment of the word however at the start of sentences

line 490: replace works against with prevents or counteracts

Figure 6: In the legend, correct), and right side, replace the abbreviated chemical structure of allicin

containing A in place of the allyl group with the proper chemical structure

Thank you for submitting your manuscript entitled "Genetic and molecular characterization of a multi-component resistance mechanism of a pseudomonad against allicin". Your work has now been peer-reviewed by two experts and I attach their reports below.

As you will see, the reviewers appreciate your data and only have minor suggestions for improvement. We would thus like to invite you to submit a revised version of your manuscript to us. While responding to the specific concerns of the two reviewer, please also pay attention to the following:

We have re-formatted the MS as required by the Journal

Reviewer #1 (Comments to the Authors (Required)):

This manuscript provides genetic and molecular characterization of the allicin resistance mechanism of the garlic-adapted bacterium *Pseudomonas fluorescens*. There is a high degree of novelty in this well-written, experimentally solid manuscript, which with minor changes merits publication as is.

Minor errors:

1) mis-use of semicolons where commas are needed (line 50 after close-parenthesis and line 490)

corrected

2) The statement on lines 450-453 is incorrect "Although onion does not produce allicin, it has many sulfur containing redox-active compounds which may be involved in defense (Imai et al. 2002), for example diallyl tetrasulfane, which was recently shown to S-thioallylate a similar spectrum of

proteins to allicin in *Bacillus subtilis* (Chi et al. 2019)." While onion does contain various sulfur compounds, there are no allyl compounds present so "diallyl tetrasulfane" cannot be correct.

An embarrassing error, thank you for spotting it. We have revised this statement.

3) The referencing style is inconsistent with some titles containing all first letters capitalized and some not. Shouldn't all botanical terms in titles be italicized as is done in the manuscript text?

Latin binomials have been italicized throughout, but the '1st letter capitals vs. lower case first letters' reflects the different citation formats in the original Journal citations and we do not feel at liberty to change that.

Apart from these minor errors, this is an excellent manuscript meriting publication without change.

Reviewer #2 (Comments to the Authors (Required)):

The manuscript by Borlinghaus et al. describes a comprehensive study of a recently discovered pseudomonad strain resistant to the highly aggressive redox modulator allicin. The authors demonstrate that this resistance is most likely acquired by horizontal gene transfer and results from the interplay of several proteins and enzymes, of which about half are redox related. Overall this is a very interesting manuscript, nicely written and presented, and of considerable significance for the field of redox biology.

Some parts, such as the section on pages 10 and 11, are very exhaustive. Then again, I am not an expert in gene transfer and such details may be of interest to colleagues active in this field.

I have spotted a few very minor mistakes which may be corrected as part of a minor revision. These are:

line 48: replace protein with proteins

done

line 78: replace work with study

done

line 384: replace lab with laboratory

done

line 444: replace lead with leads

done

line 452: check if correct as onions tend to contain the propyl- rather than allyl-derivatives

An embarrassing error, thank you for spotting it. We have revised this statement.

line 453 and 457, and thereafter: Repetitive employment of the word however at the start of sentences

We have revised this section to avoid repetiveness.

line 490: replace works against with prevents or counteracts

done

Figure 6: In the legend, correct ,), and right side, replace the abbreviated chemical structure of allicin containing A in place of the allyl group with the proper chemical structure

done

March 16, 2020

RE: Life Science Alliance Manuscript #LSA-2020-00670-TR

Prof. Alan John Slusarenko
RWTH Aachen University
Plant Physiology (Bio3)
Worringerweg 1
Aachen 52056
Germany

Dear Dr. Slusarenko,

Thank you for submitting your Research Article entitled "Genetic and molecular characterization of multi-component resistance of *Pseudomonas* against allicin". I appreciate the introduced changes and it is a pleasure to let you know that your manuscript is now accepted for publication in Life Science Alliance. Congratulations on this interesting work.

DISTRIBUTION OF MATERIALS:

Again, congratulations on a very nice paper. I hope you found the review process to be constructive and are pleased with how the manuscript was handled editorially. We look forward to future exciting submissions from your lab.

Sincerely,

Andrea Leibfried, PhD
Executive Editor
Life Science Alliance
Meyerohofstr. 1
69117 Heidelberg, Germany
t +49 6221 8891 502
e a.leibfried@life-science-alliance.org
www.life-science-alliance.org